# Synthetic CT Generation from MRI Using Improved DualGAN

**Denis Prokopenko**[1,2]        DENIS.PROKOPENKO@SKOLTECH.RU

**Joël Valentin Stadelmann**[2]        JOEL.STADELMANN@PHILIPS.COM

**Heinrich Schulz**[3]        HEINRICH.SCHULZ@PHILIPS.COM

**Steffen Renisch**[3]        STEFFEN.RENISCH@PHILIPS.COM

**Dmitry V. Dylov**[1]        D.DYLOV@SKOLTECH.RU

[1] *Skolkovo Institute of Science and Technology, Moscow, Russian Federation*

[2] *Philips Innovation Labs RUS, Moscow, Russian Federation*

[3] *Philips GmbH Innovative Technologies, Hamburg, Germany*

## Abstract

Synthetic CT image generation from MRI scan is necessary to create radiotherapy plans without the need of co-registered MRI and CT scans. The chosen baseline adversarial model with cycle consistency permits unpaired image-to-image translation. Perceptual loss function term and coordinate convolutional layer were added to improve the quality of translated images. The proposed architecture was tested on paired MRI-CT dataset, where the synthetic CTs were compared to corresponding original CT images. The MAE between the synthetic CT images and the real CT scans is 61 HU computed inside of the true CTs body shape.

**Keywords:** Machine Learning, Deep Learning, Radiology, Computed Tomography, Magnetic Resonance Imaging.

## 1. Introduction

Radiotherapy (RT) requires a personalised preparation of the treatment plan and, especially, a pre-treatment assessment of the radiation dose. The method is based on two examinations: magnetic resonance imaging (MRI) and computed tomography (CT). The MRI helps to locate the tumour and to outline its shape due to superior soft tissue contrast (Karlsson et al., 2009). The CT scan helps to obtain an attenuation map of a body part from which the radiation plan is derived. Following the MRI and CT procedures, their relative alignment, and dose calculation, the patient then undergoes the RT (Coy and Kennelly, 1980; Khuntia et al., 2006). However, an additional bias appears due to shifting errors between body alignments in MRI and CT devices. In the modern approach of synthetic CT generation, a patient has to participate only in an MRI procedure, with the advantage of having an error-free registration between the CT and the MRI volumes (Jonsson et al., 2010).

This paper focuses on the unpaired GAN-based approach to the MRI-to-CT image translation. Our contributions include the upgrade of cycle consistent model with an additional loss function term: the perceptual loss function term that enables a comparison of the high-level image representations obtained with the VGG-16 pre-trained model (Johnson et al., 2016). We also include a coordinate convolutional layer (Liu et al., 2018), which

helps to localise spatially convolutional properties (Hofmann et al., 2008). The proposed architecture was trained on the brain data with a set of almost overlapping MRI and CT scans. We present meaningful results of synthetic CT image generation.

## 2. Methods

**The DualGAN** architecture (Yi et al., 2017) consists of two image generators, which form a cycle, and two discriminators, see Figure 1. The first generator $G_{MRI \to CT}$ is trained to translate MRI images to CT images; the second generator $G_{CT \to MRI}$ translates images from CT to MRI domain. The cycle allows comparing the reconstructed image and original to evaluate the quality of generators without the need in paired data.

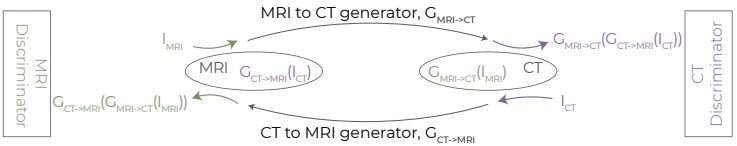

Figure 1: DualGAN architecture.

**The perceptual loss function (VGG)** builds on the idea of a feature matching, where high-level representations of two images are compared by mean squared error. Although extracted features by pre-trained VGG-16 are not optimised for tomographic images, their use is still relevant because the features extracted in an identical way for both compared images.

**The coordinate convolutional layer (CC)** allows taking into account the spatial information of the images by concatenating two additional $x$ and $y$ coordinates slices with the tensor representation of the image. The coordinate convolutional layer in application to the MRI-to-CT translation helps to distinguish black pixels of MRI image, which could represent either a bone or air.

## 3. Experiments

Three medical datasets were obtained to train and test the considered methods of unpaired MRI to CT translation. Each set was divided into a train and test parts in a 7:3 ratio. The volumes were normalised and preprocessed.

The CPTAC Phase 3 dataset includes the MRI T1-weighted images of 7 patients. Each 3D volume of a patient contains $22 - 24$ slices in the axial anatomical plane comprised between the lower nose and the top of the head region (CPTAC, 2018; Clark et al., 2013).

The head-and-neck cancer dataset consists of CT scans of 61 patients. 3D volumes include 61 - 94 slices from the shoulders, neck and lower part of a head up to the eyes (Vallières et al., 2017; Vallières et al., 2017).

The private dataset contained CT images, masks and paired MRI T1-weighted images of 10 patients. The volumes consist of $66-137$ slices in the axial plane, which are comprised between the teeth and the top of the head region. Train part was used in an unpaired way.

All models were trained separately on CPTAC with Head-and-neck cancer datasets together and the private dataset. The qualitative results and quantitative evaluation of the performance were obtained on the test part of the private dataset.

In this work, four different architectures were considered and compared: DualGAN, DualGAN with the coordinate convolutional layer (DualGAN, CC), DualGAN with the perceptual loss function term (DualGAN, VGG) and DualGAN with perceptual loss function term and coordinate convolutional layer (DualGAN, VGG, CC). All of them work with three-channel images allowing the application of VGG network.

Performance evaluation was calculated using one channel image representations. The quantitative results of different translation configurations can be seen in Table 1, while qualitative results are presented in Figure 2, which shows by column the real MRI slices, the synthetic CT images, the real paired CT slices and the difference between synthetic and original CTs.

Table 1: Performance comparison of different MRI to CT translation configurations.

| Configuration | MAE, HU ↓ | PSNR, dB ↑ | SSIM ↑ |
|---|---|---|---|
| DualGAN | $62.95 \pm 1.17$ | $16.82 \pm 0.83$ | $0.79 \pm 0.02$ |
| DualGAN, CC | $63.27 \pm 2.00$ | $16.96 \pm 0.71$ | $0.78 \pm 0.03$ |
| DualGAN, VGG | $66.52 \pm 3.74$ | $16.74 \pm 1.08$ | $0.78 \pm 0.03$ |
| DualGAN, VGG, CC | $60.83 \pm 2.20$ | $17.21 \pm 1.00$ | $0.80 \pm 0.03$ |

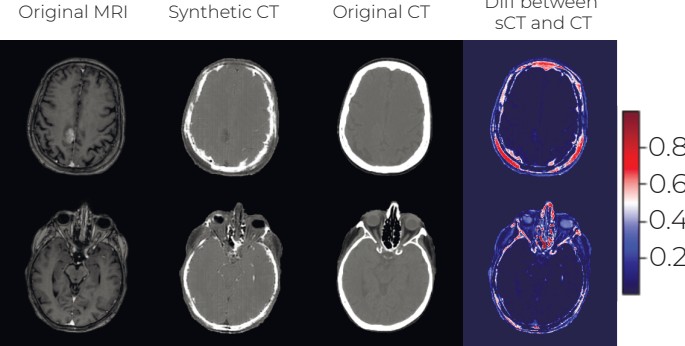

Figure 2: Tanslation made by DualGAN with perceptual loss function term and coordinate convolutional layer.

## 4. Conclusions

The presented model achieves MAE of 61 HU and SSIM of 0.8, which compares favourably to scientific literature (Wolterink et al., 2017). The translation architecture transforms the initial image retaining the structural information. While the DualGAN with the perceptual loss function term and the DualGAN with the coordinate convolutional layer are not themselves superior to the DualGAN model, their triad combination showed meaningful improvement. Besides, our model works with the unpaired images, and the visual examination confirmed that the use of the perceptual loss term and the coordinate convolutional layer enhances the appearance of the resulting images. The error of translation increases with the complexity of the inner structures, for instance, in the nose and teeth regions.

## 5. Acknowledgement

Data used in this publication were generated by the National Cancer Institute Clinical Proteomic Tumor Analysis Consortium (CPTAC).

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
