# OpenReview forum: "Synthetic CT Generation from MRI Using Improved DualGAN"
_MIDL.io/2019/Conference/Abstract — MIDL Abstract 2019_

### Official Review · AnonReviewer1 · 2019-04-29
**Interesting result in an important novel application**

**Rating:** 3
**Confidence:** 3

**Review:**

This is an attempt to create CT from MRI slices to allow to omit the need to acquire a CT scan for radiotherapy planning in situations where an MR scan is already available. A standard cycleGAN is used, with two novelties aimed to improve the results (a perceptual loss function and a coordconv layer). The results are visually not very convincing, the CT slices obviously do not look like CT at all at many places, and this may be because the number of scans used for training is too small. Another weakness is the fact that a 2D approach is used while this is 3D data. Still, the authors claim to outperform previously published work in this area. This, and the fact that this is an interesting novel application make this work an interesting contribution to MIDL.

---

### Official Review · AnonReviewer2 · 2019-04-30
**Preliminary results showing feasibility of dual GAN with modifications for synthetic CT generation in Head and Neck**

**Rating:** 3
**Confidence:** 3

**Review:**

The paper presents very preliminary results (I say preliminary due to the quality of the results - please see more below and my suggestions to possibly improve experiments) for synthetic CT generation from for head and neck from T1-weighted MRI.
Pros:
1. Clinical: This is a very relevant and important topic for MR-guided radiation therapy
2. Technical: Incremental innovation for improving MR to CT image translation by employing a well-known loss, perceptual loss, and an approach to potentially preserve spatial information using coordinate convolution.
3. Cons:
1. Technical: The technical details do not show how the method is useful other than making incremental improvements. Providing at least a bit more insights on why these improvements helped in the discussion and the limitation of the method is necessary. For instance, why is the perceptual loss helping? This loss could potentially help to capture some of the edge characteristics in the CT, but keep in mind that you go from a high-dimensional representation (MR which has much better soft tissue resolution) to CT (which has much poorer soft-tissue resolution). So how is perceptual loss helping? The same goes for the coordinate convolution. Authors motivate this to say that it helps to get better differentiation between air and bone. But in head and neck, these two often occur adjacently, ex. sinuses. So how does this really help?
2. Results: While the results look competitive to the published results, there does not seem to be significant difference between the different methods from the means. Adding standard deviation is necessary. Also, why is DualGAN+CC worse? No insight on that is provided.
3. Results visual: The synthetic CT results for the bone regions look very poor. Why is that? Even Pix2Pix can often get better results than this. There is a clear blurring and diffuse appearance of bones. This is actually big problem for treatment planning.
4. Minor results: Synthetic CT results are shown for which method? You show results of Difference image almost as a binary image. Colormap/heat map with a colorbar would be better way of showing this.
5. Minor: Lots of English and grammatical errors.

---

### Decision · Program_Chairs · 2019-05-06
**Acceptance Decision**

Accept